# Biopsychosocial factors associated with symptom severity in the overlap of non-erosive reflux disease and epigastric pain syndrome: A multicenter cross-sectional study

**Mi Lv[1], Hui Che[1], Jiayan Hu[2], Wenxi Yu[1], Zhaoxia Liu[3], Xiaolin Zhou[4¤], Binduo Zhou[5], Jinyi Xie[1], Fengyun Wang** [1]ORCID[1]*

1 Xiyuan Hospital, China Academy of Chinese Medical Sciences, Beijing, China, 2 Dongfang Hospital, Beijing University of Chinese Medicine, Beijing, China, 3 The First Affiliated Hospital, Heilongjiang University of Chinese Medicine, Harbin, Heilongjiang, China, 4 Yueyang Hospital of Integrated Traditional Chinese and Western Medicine, Shanghai University of Traditional Chinese Medicine, Shanghai, China, 5 Liuzhou Traditional Chinese Medical Hospital (Liuzhou Zhuang Medicine Hospital), Liuzhou, Guangxi, China

¤ Current address: The First Affiliated Hospital of Guangxi University of Chinese Medicine, Guangxi, China
* 18810631761@163.com

## Abstract

### Background

The overlap between non-erosive reflux disease (NERD) and epigastric pain syndrome (EPS, a subtype of functional dyspepsia) is common, yet its associated factors remain poorly defined. We aimed to identify factors associated with symptom severity in NERD-EPS overlap, focusing on psychosocial and somatic factors.

### Methods

In this multicenter cross-sectional study, 800 patients meeting Rome IV criteria for NERD-EPS overlap were enrolled. Standardized questionnaires assessed gastrointestinal symptoms (GSRS), somatic symptoms (PHQ-15), anxiety/depression (PHQ-4), and sleep quality (SRSS). Multivariable regression models identified factors independently associated with GSRS scores, adjusted for demographics and clinical covariates. Interaction terms were tested to assess whether the association between one factor and GSRS scores varied across different levels of another factor.

### Results

Of the 800 patients, 67% were female, and the mean age was (44.50 ± 14.43) years. 67% had mild or more sleep problems, and 47% had anxiety or depression. Somatic symptoms (PHQ-15) showed the strongest association with GSRS scores ($\beta = 0.617$, $P < 0.001$), followed by poor sleep quality (SRSS; $\beta = 0.115$, $P < 0.001$) and anxiety/depression (PHQ-4; $\beta = 0.069$, $P = 0.026$). Urban residence ($\beta = 0.071$) and mixed labor type ($\beta = -0.066$) were also independently associated with symptom burden.

**Data availability statement:** All relevant data are within the manuscript and its Supporting Information files.

**Funding:** This study was jointly supported by: the National Key Specialty of Traditional Chinese Medicine (Spleen and Stomach Diseases, 0500004), the Innovation Project of China Academy of Chinese Medical Sciences (CI2021A01001), and the Youth Science Foundation Project of the National Natural Science Foundation of China (82205104, 82104850). The funders had no role in study design, data collection and analysis, decision to publish, or preparation of the manuscript.

**Competing interests:** The authors have no competing interests to declare that are relevant to the content of this article.

## Conclusion

Somatic symptoms, psychological distress, and sleep disturbances were the factors most strongly associated with symptom severity in NERD-EPS overlap, with additional contributions from younger age, male sex, and urban residence. Our findings advocate for integrated biopsychosocial interventions to alleviate symptom burden in this population.

---

## Introduction

Non-erosive reflux disease (NERD) is an upper gastrointestinal disorder characterized by reflux of gastroduodenal contents into the esophagus causing symptoms of reflux/acid reflux and heartburn, but no endoscopic damage to the esophageal mucosa, is the most common subtype of Gastroesophageal Reflux Disease (GERD), with a percentage of GERD that can be as high as 79.4% [1,2]. Epigastric pain syndrome (EPS) is diagnosed by a disease duration of more than 6 months, with symptoms of epigastric pain or burning sensation in the epigastrium evident in the last 3 months, and no organic changes in the gastric mucosa, such as erosions or ulcers, are seen endoscopically, is a subtype of Functional Dyspepsia (FD), which is commonly seen in FD in more than 40% of cases [2–4]. In the outpatient system, patients diagnosed with NERD frequently have coexisting EPS symptoms, and patients presenting to the clinic with a predominant complaint of EPS symptoms may also qualify for a diagnosis of NERD. In a study by Noh YW et al, EPS symptoms were also found to be more common than PDS symptoms in patients with NERD (68.9% vs. 48.6%), in addition to epigastric pain which was considered the most uncomfortable FD symptom by about 30% of patients with NERD [5,6].

Current evidence indicates that GERD/NERD frequently coexists with FD in clinical practice. While multiple studies have demonstrated that this overlap syndrome exacerbates patients' gastrointestinal symptoms, psychological distress, and sleep quality, the specific clinical characteristics of NERD-EPS overlap remain poorly characterized [7–9]. Furthermore, the correlates of gastrointestinal symptom severity in NERD-EPS overlap remain to be elucidated.

While the aforementioned studies from Asian cohorts have provided valuable insights into the overlap of GERD and FD, the specific NERD-EPS phenotype remains less explored on a global scale. Epidemiological data from the Rome Foundation Global Study confirm the high prevalence and global burden of GERD-FD overlap, yet detailed characterization of the NERD-EPS subtype, particularly its biopsychosocial determinants, is scarce [4]. For instance, a study in a European population has reported significant overlap between GERD and FD, highlighting shared pathophysiological pathways such as visceral hypersensitivity [9]. However, whether the clinical profile and key associated factors identified in Asian cohorts, such as the prominent role of somatic symptoms, are consistent across different ethnic and cultural backgrounds remains unclear. This study, therefore, aims not only

to characterize the NERD-EPS overlap within a Chinese population but also to contribute data that is crucial for future cross-cultural comparisons and the development of universally applicable management strategies.

Specifically, our study aimed to elucidate the characteristics of clinical symptom distribution in patients with NERD-EPS overlap through a multicenter cross-sectional study and to assess their clinical profiles using the Patient Health Questionnaire-4 (PHQ-4) and the Patient Health Questionnaire-15 (PHQ-15) [10–13]. The Gastrointestinal Symptom Rating Scale (GSRS) and Self-Rating Scale of Sleep (SRSS) were further employed to assess the factors affecting GSRS scores in these patients [14–17].

## Methods

### Study design

This multicenter study was conducted from March 5, 2022, to October 27, 2024, which involved gastroenterology departments at four tertiary Traditional Chinese Medicine hospitals across four Chinese provinces. These sites were strategically selected to provide balanced geographical representation of both northern and southern China. All participants were fully informed of the study procedures and provided written informed consent prior to enrollment. The study protocol was approved by the Institutional Review Boards of all participating hospitals (Approval Nos: 2022XLA041−1, 2022-039-01, HZYLLKY202200701, 2022-KY-HX-12−01). This research involved neither minors nor the use of medical records/archived samples. All collected data were anonymized to protect participant confidentiality.

### Study population

Eight hundred patients aged 18–75 years who met the diagnosis of NERD with EPS and obtained informed consent were enrolled. The presence of gastrointestinal alarm symptoms included in the Rome IV Diagnostic Scale for functional gastrointestinal diseases (FGIDs) (e.g., black stools, abdominal masses, and a family history of gastrointestinal tumors); the presence of severe other systemic and infectious diseases; and the inability to cooperate with the questionnaire were excluded [3]. All enrolled patients underwent endoscopy to rule out erosive disease and ulcers. Helicobacter pylori status was assessed, and current medication use (including acid-suppressants and antidepressants) was documented for descriptive purposes.

### Definitions

The diagnosis of NERD was established according to the 2020 Chinese Expert Consensus on Gastroesophageal Reflux Disease, which requires: [18] (a) presence of typical heartburn and reflux symptoms, plus either (b) a positive proton pump inhibitor (PPI) test, (c) endoscopic confirmation of normal esophageal mucosa with exclusion of reflux esophagitis (RE), Barrett's esophagus (BE) and other upper gastrointestinal pathologies, or (d) abnormal findings on 24-hour esophageal pH-impedance monitoring.

EPS was diagnosed based on Rome IV criteria for FGIDs, now also classified under the modern nomenclature of disorders of gut–brain interaction (DGBI), requiring: (a) epigastric pain or (b) burning sensation severe enough to impair daily activities, occurring ≥ 1 day per week. Patients had to fulfill at least one criterion with symptom persistence for ≥6 months, including active symptoms meeting diagnostic thresholds during the preceding 3 months [3].

### Questionnaire design

In addition to collecting demographic and clinical characteristics (including gender, age, body mass index (BMI), disease duration of NERD/EPS, occupation type, educational attainment, marital status, residential area, smoking/alcohol consumption history, history of inflammatory bowel disease, and history of abdominal surgery), standardized instruments were administered: PHQ-4 for anxiety/depression screening, PHQ-15 for somatic symptom burden, SRSS for sleep

quality assessment, GSRS for gastrointestinal symptom severity. Healthcare utilization was quantified through clinic visit frequency during the preceding 3-month period. Occupational categories were defined as follows: Manual labor: Jobs primarily involving physical effort (e.g., construction, farming). Mental labor: Jobs primarily involving cognitive tasks (e.g., office work, management). Mixed labor: Jobs requiring a comparable combination of physical and cognitive duties (e.g., healthcare workers, technicians, retail supervisors).

### Anxiety and depression

The PHQ-4 is a validated ultra-brief screening instrument comprising two subscales: PHQ-2 for core depressive symptoms and GAD-2 for generalized anxiety symptoms. PHQ-2 (depression) and GAD-2 (generalized anxiety) [10,11]. Each item is rated on a 4-point Likert scale: 0 = "Not at all", 1 = "Several days", 2 = "More than half the days", 3 = "Nearly every day". Total scores range from 0 to 12, with elevated scores indicating greater symptom severity. This study analyzed both continuous total scores and categorical classifications (asymptomatic [0–2], mild [3–5], moderate [6–8], severe [9–12]) in accordance with standard clinical cutoffs.

### Somatic symptoms

The PHQ-15 is a validated self-report instrument designed to quantify somatic symptom burden through assessment of 15 prevalent physical complaints [12,13]. Symptoms are rated on a standardized 3-point scale: 0 = "No distress", 1 = "Mild distress", 2 = "Severe distress". Total scores range from 0 to 30, with established clinical thresholds: Minimal (0–4), Low (5–9), Medium (10–14), High (≥15). Demonstrating excellent reliability in FGIDs, the PHQ-15 has become the gold standard for psychosomatic evaluation in clinical research. Our analysis incorporated both dimensional (continuous total scores) and categorical (severity-stratified) approaches.

### Gastrointestinal symptoms

The GSRS is a validated multidimensional instrument that evaluates 15 symptom items across five clinically relevant domains: (1) abdominal pain, (2) reflux syndrome, (3) diarrhea, (4) constipation, and (5) dyspepsia [14,15]. Symptoms are rated on a 4-point severity scale: 1 = "No discomfort (asymptomatic)", 2 = Mild discomfort (noticeable but not limiting daily activities), 3 = "Moderate discomfort (interfering with but not preventing daily activities)", 4 = "Severe discomfort (incapacitating, preventing normal activities)". Total scores range from 15 to 60, with higher scores indicating greater gastrointestinal symptom severity. The GSRS has demonstrated excellent reliability and validity in FGIDs research. For the current analysis, we utilized the continuous total score to maximize statistical power.

### Sleep quality

The SRSS is a validated 10-item instrument assessing four critical sleep domains: [16,17] (1) sleep initiation, (2) sleep maintenance, (3) early morning awakening, and (4) daytime dysfunction. Items are scored on a 5-point Likert scale with standardized anchors: 1 = "Never" (no sleep disturbance), 2 = "Occasionally" (<1 night/week), 3 = "Sometimes" (1–2 nights/week), 4 = "Often" (3–4 nights/week), 5 = "Always" (≥5 nights/week). Total scores range from 10 to 50, with established clinical thresholds: Normal sleep (10–19), Mild disturbance (20–29), Moderate disturbance (30–39), Severe disturbance (40–50). The SRSS was analyzed using both continuous scores and categorical classifications (normal/mild/moderate/severe) to capture comprehensive sleep profiles.

### Standardized data collection protocol

To ensure methodological rigor, we implemented a comprehensive quality control protocol beginning with questionnaire validation through pilot testing (n = 30) and cognitive debriefing with 5 gastroenterologists to establish content validity. All

research assistants completed a 12-hour certification program including standardized administration procedures, mental status evaluation training by board-certified psychiatrists, and final competency assessments. Under rigorous quality control protocols, trained research staff administered all questionnaires while maintaining standardized assistance limited to: (1) clarifying ambiguous terms, (2) explaining item intent without leading responses, and (3) ensuring complete understanding of response options, with all procedures safeguarding participant confidentiality and autonomous decision-making as per HIPAA guidelines [19].

## Statistical analysis

Continuous variables were presented as mean ± standard deviation, while categorical variables were summarized using frequencies and percentages. Nonparametric analyses were employed as follows: (1) Mann-Whitney U or Kruskal-Wallis tests for associations between categorical variables and GSRS scores, with post-hoc pairwise comparisons when $P < 0.05$; (2) Spearman's rank correlation for continuous variable associations. Variables showing significant correlations ($P < 0.05$) were eligible for inclusion in multivariable modeling.

## Regression modeling strategies

Given the substantial number of candidate variables (n = 20), we adopted a two-stage analytical approach: [20] (1) Full-model specification: All prespecified variables were entered simultaneously without preliminary screening; (2) Refined modeling: Variables significant in univariate analyses were retained for final models. Categorical variables were dummy-coded (binary: 0/1; multinomial: k-1 variables). Interaction terms (e.g., A × B) were incorporated based on theoretical plausibility and univariate results [21].

## Model diagnostics

We evaluated model quality through: Goodness-of-fit: Adjusted R² and Akaike Information Criterion (AIC); Multicollinearity: Variance inflation factors (VIF > 5 indicating severe collinearity); Residual analysis: Q-Q plots and standardized residual plots for heteroscedasticity; Robustness checks: Ridge regression and variable transformations when assumptions were violated [22].

All analyses were performed using IBM SPSS Statistics (version 26.0; IBM Corp., Armonk, NY, USA).

## Subgroup and exploratory analyses

To explore potential heterogeneity and identify clinically relevant patient profiles, we conducted post-hoc subgroup analyses. The study population was stratified based on:

Sex (Male vs. Female), somatic symptom severity based on PHQ-15 scores, categorized as Minimal/Low (0–9), Medium/High (≥10) according to established clinical cut-offs [13]. Sleep quality based on SRSS scores, categorized as Normal/Mild (10–29) and Moderate/Severe (30–50) disturbance based on previously validated thresholds [16,17].

We then examined the differences in gastrointestinal symptom severity (GSRS scores) across these subgroups using non-parametric tests (Kruskal-Wallis H or Mann-Whitney U test). Post-hoc pairwise comparisons with Bonferroni correction were performed when overall $P < 0.05$.

All the above operations were performed in SPSS version 26.0

## Results

### Participants characteristics

Table 1 presents the clinical and demographic characteristics of the 800 enrolled patients, showing a male-to-female ratio of 3:5 with a mean age of 44.50 ± 14.43 years. The Gastrointestinal Symptom Rating Scale (GSRS) revealed mean

**Table 1. Multicenter cohort characteristics: Demographic, lifestyle and psychometric profiles in NERD-EPS overlap (N = 800).**

| Variables | N (%) | Variables | N (%) |
|---|---|---|---|
| Age | 44.50 ± 14.43 | Yes | 101 (12.63%) |
| Sex | | No | 699 (87.38%) |
| Male | 300 (33.33%) | History of abdominal surgery | |
| Female | 500 (66.67%) | Yes | 74 (9.25%) |
| BMI | 23.17 ± 4.80 | No | 726 (90.75%) |
| Disease duration | | Digestive clinic visits in past 3 months | |
| NERD | 32.18 ± 58.74 | 1-2 times | 461 (57.63%) |
| EPS | 23.94 ± 38.45 | 3-4 times | 168 (21.00%) |
| Occupational category | | 5-6 times | 161 (20.13%) |
| Manual labor | 191 (23.88%) | PHQ-4 score | |
| Mental labor | 417 (52.13%) | Asymptomatic (0–2) | 421 (52.63%) |
| Mixed labor | 192 (24.00%) | Mild (3–5) | 242 (30.25%) |
| Education level | | Moderate (6–8) | 109 (13.63%) |
| Primary school | 41 (5.13%) | Severe (9–12) | 28 (3.50%) |
| Secondary school | 281 (35.13%) | PHQ-15 score | |
| Tertiary education | 478 (59.75%) | Asymptomatic (0–4) | 205 (25.63%) |
| Primary residence | | Mild (5–9) | 278 (34.75%) |
| Urban | 695 (86.88%) | Moderate (10–14) | 204 (25.50%) |
| Rural | 105 (13.13%) | Severe (15–30) | 113 (14.13%) |
| Marital status | | SRSS score | |
| Yes | 597 (74.63%) | Asymptomatic (10–19) | 263 (32.88%) |
| No | 203 (25.38%) | Mild (20–29) | 357 (44.63%) |
| Smoking history | | Moderate (30–39) | 158 (19.75%) |
| Yes | 135 (16.88%) | Severe (40–50) | 22 (2.75%) |
| No | 665 (83.13%) | GSRS score | |
| Drinking history | | Mild (15–30) | 336 (42.00%) |
| Yes | 319 (39.88%) | Moderate (31–45) | 308 (38.50%) |
| No | 481 (60.13%) | Moderately Severe (46–60) | 126 (15.75%) |
| History of intestinal inflammation | | Severe (> 60) | 30 (3.75%) |

domain scores of 5.75 ± 2.79 for reflux, 7.39 ± 3.43 for abdominal pain, 10.37 ± 4.48 for dyspepsia, 5.61 ± 3.31 for constipation, and 6.02 ± 3.67 for diarrhea. Clinically significant comorbidities were prevalent, including mild-to-severe sleep disturbances in 67% (n = 537) of patients, anxiety/depression symptoms in 47% (n = 379), and frequent gastroenterology visits (≥3 in 3 months) in 41% (n = 329) of cases, collectively indicating substantial disease burden in this population.

## One-way analysis of variance

Univariate analysis using nonparametric methods (Mann-Whitney U, Kruskal-Wallis H, or Spearman's ρ as appropriate) identified several variables significantly associated with gastrointestinal symptom severity (GSRS scores). Key clinical factors included disease duration of both NERD ($\rho = 0.139$, $P < 0.001$) and EPS ($\rho = 0.164$, $P < 0.001$), along with occupational category (H = 31.526, $P < 0.001$). Behavioral factors showing significant associations were smoking history (U = 39,967.500, $P = 0.044$) and histories of intestinal inflammation (U = 27,479.500, $P < 0.001$) or abdominal surgery (U = 22,882.000, $P < 0.036$). Psychological and sleep-related measures demonstrated particularly strong correlations: PHQ-4 for anxiety/depression ($\rho = 0.474$, $P < 0.001$), PHQ-15 somatic symptoms ($\rho = 0.722$, $P < 0.001$), and SRSS sleep

scores ($\rho=0.513$, $P<0.001$). Post-hoc analysis of occupational categories revealed that mixed labor types had significantly lower GSRS scores (rank mean = 326.02) versus manual (455.77) or mental labor (409.48; overall $P<0.001$). Demographic factors (age, gender, BMI, education, residence, marital status) showed no significant associations (all $P>0.05$). See Table 2 for details.

## Multivariable linear regression analysis ($P<0.05$ for entry)

Variables showing statistical significance in univariate analysis ($P<0.05$) were entered into a multiple linear regression model (Table 3). The final model demonstrated strong explanatory power (adjusted $R^2=0.534$, F = 92.59, $P<0.001$) and all variables had VIFs < 2.5. PHQ-15 somatic symptoms ($\beta=0.606$, B = 1.415, $P<0.001$) showed the strongest association, followed by SRSS sleep quality ($\beta=0.093$, $P=0.003$) and PHQ-4 anxiety/depressive symptoms ($\beta=0.081$, $P=0.009$). EPS disease duration showed a positive association ($\beta=0.086$, $P=0.011$), whereas mixed labor type exhibited a protective effect ($\beta=-0.070$, $P=0.007$). Model diagnostics confirmed the absence of multicollinearity (all VIF < 2.1), and standardized residuals (mean = 0, SD = 0.994) approximated a normal distribution.

## Fully adjusted multivariable linear regression model

All prespecified variables were included in the forced-entry multiple linear regression model. The final model demonstrated robust explanatory power (adjusted $R^2=0.542$, F = 56.687, $P<0.001$). PHQ-15 score ($\beta=0.617$, B = 1.440, $P<0.001$)

**Table 2. Univariate associations between clinical variables and gastrointestinal symptom severity (GSRS scores).**

| Variables | U/H/ρ | P | Variables | U/H/ρ | P |
|---|---|---|---|---|---|
| Age | 0.060[c] | 0.089 | Drinking history | 74111.000[a] | 0.415 |
| Sex | 73604.000[a] | 0.659 | Smoking history | 39967.500[a] | 0.044 |
| BMI | −0.021[c] | 0.560 | History of intestinal inflammation | 27479.500[a] | 0.000 |
| Disease duration of NERD | 0.139[c] | 0.000 | History of abdominal surgery | 22882.000[a] | 0.036 |
| Disease duration of EPS | 0.164[c] | 0.000 | Education level | 0.582[b] (df = 2) | 0.747 |
| Occupational category | 31.526[b] (df = 2) | 0.000 | PHQ – 4 score | 0.474[c] | 0.000 |
| Primary residence | 34078.500[a] | 0.275 | PHQ – 15 score | 0.722[c] | 0.000 |
| Marital status | 58248.000[a] | 0.409 | SRSS score | 0.513[c] | 0.000 |

Note: [a] Mann-Whitney U, [b] Kruskal-Wallis H, [c] Spearman's ρ.

**Table 3. Multivariable linear regression analysis of factors associated with GSRS scores.**

| Variable | B | β | 95% CI | P |
|---|---|---|---|---|
| (Constant) | 21.984 | – | [15.591, 28.378] | 0.000 |
| Disease duration of NERD | − 0.006 | − 0.025 | [- 0.020, 0.009] | 0.458 |
| Disease duration of EPS | 0.029 | 0.086 | [0.007, 0.051] | 0.011 |
| Smoking history | − 1.585 | − 0.046 | [- 3.233, 0.063] | 0.059 |
| History of intestinal inflammation | 0.125 | 0.003 | [- 1.797, 2.047] | 0.899 |
| History of abdominal surgery | − 0.570 | − 0.013 | [- 2.779, 1.639] | 0.613 |
| PHQ – 4 score | 0.361 | 0.081 | [0.090, 0.632] | 0.009 |
| PHQ – 15 score | 1.415 | 0.606 | [1.255, 1.574] | 0.000 |
| SRSS score | 0.161 | 0.093 | [0.055, 0.267] | 0.003 |
| Manual Labor | − 0.770 | − 0.026 | [- 2.338, 0.797] | 0.335 |
| Mixed labor | − 2.101 | − 0.070 | [- 3.621, - 0.581] | 0.007 |

remained the variable most strongly associated with GSRS scores. Additional variables independently associated with GSRS scores included age (β = −0.076, *P* = 0.041), female sex (β = −0.070, *P* = 0.018), and urban residence (β = 0.071, *P* = 0.007), although these variables were not significant in univariate analysis.

Model diagnostics confirmed the absence of multicollinearity (all variance inflation factors [VIFs] < 2.5). Standardized residuals (mean = 0, SD = 0.989) followed an approximately normal distribution, supporting model validity. Details are shown in Table 4.

It is noteworthy that several demographic variables—namely younger age, female sex, and urban residence—were identified as statistically significant independent factors associated with higher GSRS scores in the fully adjusted model, despite not showing significance in the initial univariate analyses. This phenomenon can be attributed to the effect of statistical adjustment. In univariate analysis, the true effect of these demographic factors may have been masked or confounded by their relationships with other stronger correlates (e.g., somatic symptoms, psychological distress). The multivariable model, by simultaneously accounting for all these covariates, isolates the unique contribution of each variable, thereby revealing their independent associations with symptom severity that were otherwise obscured.

## Gastrointestinal symptom severity across clinical subgroups

Stratified analyses revealed significant differences in gastrointestinal symptom burden (GSRS scores) across subgroups defined by somatic symptoms and sleep quality, but not by sex (Table 5). Patients with medium-to-high somatic symptom burden (PHQ-15 ≥ 10) had substantially higher median GSRS scores than those with minimal-low burden (42.0 vs. 28.0, U = 129,021.500, *P* < 0.001). Similarly, median GSRS scores were higher in patients with moderate-to-severe sleep disturbances (SRSS ≥30) compared to those with normal-mild sleep quality (41.0 vs. 31.0, U = 81,725.500, *P* < 0.001). In contrast, no statistically significant difference in GSRS scores was observed between male and female patients (median: 33.0 vs. 34.0, U = 76,396.000, *P* = 0.659).

**Table 4. Fully adjusted multivariable regression analysis of GSRS scores determinants.**

| Variable | B | β | 95% CI | *P* |
|---|---|---|---|---|
| (Constant) | 22.865 | – | [12.632, 33.097] | 0.000 |
| Age | −0.067 | −0.076 | [-0.132, -0.003] | 0.041 |
| Sex | −1.856 | −0.070 | [-3.394, -0.319] | 0.018 |
| BMI | 0.015 | 0.005 | [-0.113, 0.143] | 0.822 |
| Disease duration of NERD | −0.002 | −0.010 | [-0.017, 0.013] | 0.775 |
| Disease duration of EPS | 0.024 | 0.073 | [0.002, 0.047] | 0.031 |
| Education Level | −0.500 | −0.023 | [-1.862, 0.863] | 0.472 |
| Primary residence | 2.717 | 0.071 | [0.758, 4.675] | 0.007 |
| Marital status | −0.297 | −0.010 | [-2.096, 1.502] | 0.746 |
| Smoking history | −0.710 | −0.021 | [-2.758, 1.338] | 0.497 |
| Drinking history | 0.302 | 0.011 | [-1.187, 1.791] | 0.691 |
| History of intestinal inflammation | 0.596 | 0.015 | [-1.330, 2.522] | 0.543 |
| History of abdominal surgery | −0.923 | −0.021 | [-3.129, 1.283] | 0.412 |
| PHQ-4 score | 0.308 | 0.069 | [0.036, 0.580] | 0.026 |
| PHQ-15 score | 1.440 | 0.617 | [1.280, 1.601] | 0.000 |
| SRSS score | 0.200 | 0.115 | [0.091, 0.309] | 0.000 |
| Manual labor | −0.510 | −0.017 | [-2.371, 1.351] | 0.591 |
| Mixed labor | −1.981 | −0.066 | [-3.676, -0.286] | 0.022 |

**Table 5. Differences in gastrointestinal symptom severity (GSRS scores) across patient subgroups.**

| Variable | Subgroup | GSRS score, median | U | P |
|---|---|---|---|---|
| Sex | Male | 33 | 76396.000 | 0.659 |
| | Female | 34 | | |
| PHQ-15 score | Minimal/Low (0–9) | 28 | 129021.500 | 0.000 |
| | Medium/High (≥10) | 42 | | |
| SRSS score | Normal/Mild (10–29) | 31 | 81725.500 | 0.000 |
| | Moderate/Severe (30–50) | 41 | | |

## Clinical interpretation of key associations

To translate these statistical associations into clinically meaningful insights, we estimated the impact of these key factors on the GSRS score. The strong association with the PHQ-15 score (B = 1.440) indicates that for every 10-point increase in somatic symptom burden—a change that, for example, corresponds to a shift from the 'low' (5–9) to the 'medium' (10–14) severity category—the total GSRS score increases by 14.4 points. This represents a substantial worsening of gastrointestinal symptoms. Similarly, a 10-point increase in the SRSS score (B = 0.200), reflecting a significant deterioration in sleep quality, was associated with a 2.0-point increase in GSRS.

## Interaction test

To explore whether the associations between key variables and GSRS scores were modified by other factors, we tested several interaction terms based on theoretical plausibility and univariate results [23–31]. These included, for example, the interaction between disease duration of EPS and PHQ-4 score, as well as between PHQ-4 and SRSS scores. As shown in Table 6, none of the interaction terms reached statistical significance (all $P > 0.05$), indicating that the associations between these factors and GSRS scores were consistent across different levels of the moderating variables, with no evidence of effect modification. Note: [a] Interaction terms are from multivariate linear regression models after univariate screening, [b] Interaction terms are from full-variable multivariate linear regression models.

**Table 6. Exploratory analysis of interaction effects on GSRS scores in NERD-EPS overlap patients.**

| interaction term | B | β | 95% CI | P |
|---|---|---|---|---|
| Age x SRSS score | −0.005[b] | −0.229[b] | [-0.011, 0.001][b] | 0.082[b] |
| Sex x SRSS score | 0.018[b] | 0.024[b] | [-0.158, 0.194][b] | 0.845[b] |
| Disease duration of EPS x PHQ-4 score | 0.002[a] | 0.038[a] | [-0.002, 0.007][a] | 0.283[a] |
| | 0.002[b] | 0.030[b] | [-0.002, 0.006][b] | 0.401[b] |
| PHQ-4 score × SRSS score | 0.021[a] | 0.150[a] | [-0.004, 0.047][a] | 0.104[a] |
| | 0.019[b] | 0.136[b] | [-0.006, 0.045][b] | 0.140[b] |
| PHQ-4 score × PHQ-15 score | −0.014[a] | −0.052[a] | [-0.046, 0.019][a] | 0.414[a] |
| | −0.018[b] | −0.069[b] | [-0.050, 0.015][b] | 0.282[b] |
| Mixed labor × PHQ-4 score | 0.371[a] | 0.049[a] | [-0.142, 0.883][a] | 0.156[a] |
| | 0.318[b] | 0.042[b] | [-0.192, 0.827][b] | 0.221[b] |
| Mixed labor x SRSS score | −0.032[a] | −0.026[a] | [-0.214, 0.150][a] | 0.733[a] |
| | −0.035[b] | −0.028[b] | [-0.217, 0.148][b] | 0.709[b] |

## Discussion

While previous studies have established psychosocial factors in isolated GERD or dyspepsia cohorts [7–9], this pioneering multicenter study is the first to delineate their relative contributions specifically in the NERD-EPS overlap population. Our novel finding reveals a distinct hierarchy of effect sizes: somatic symptom burden (PHQ-15 scores, β = 0.617, $P < 0.001$) demonstrated a substantially stronger association with gastrointestinal symptom severity (GSRS scores) than sleep disturbances (SRSS scores, β = 0.115, $P < 0.001$) and anxiety/depression (PHQ-4 scores, β = 0.069, $P < 0.05$), suggesting a predominant "somatic amplification" mechanism that distinguishes this overlap syndrome from more psychologically-driven profiles in pure GERD or FD [7,9,32,33]. We further identified unique correlates including an inverse association between mixed labor type and symptom severity (β = −0.066, $P < 0.05$) and adverse associations with urban residence (β = 0.071) [34,35]. The observed association with mixed labor may reflect a more balanced work-life rhythm or lower occupational stress compared to purely manual or mental occupations.

Our subgroup analyses further clarify these associations. The stark contrast in GI symptom burden between patients with medium-to-high versus minimal-low somatic symptom burden (Median GSRS: 42.0 vs. 28.0) underscores a clear dose-response relationship, establishing a PHQ-15 score ≥10 as a practical clinical marker for severe GI distress. Similarly, significant GSRS differences across SRSS categories position sleep disturbance as a key correlate of symptom severity. Interestingly, the lack of a significant sex difference in median GSRS scores, despite its association in the multivariate model, suggests the protective effect of female sex is subtle and likely mediated through complex pathways rather than a direct reduction in raw symptom severity.

The hierarchy of biopsychosocial factors identified in our Chinese cohort—with somatic symptoms demonstrating the strongest association—provides a crucial lens through which to view the NERD-EPS overlap phenotype globally. While the predominance of somatic amplification may be more pronounced in cultural contexts that favor somatization, the central role of the brain-gut axis in this hierarchy is likely a universal mechanism. Our findings align with a growing international consensus that disorders of DGBI, including FD and GERD, are fundamentally driven by central nervous system dysregulation, leading to visceral hypersensitivity and central sensitization [32]. The strong link between PHQ-15 and GSRS scores in our study can be interpreted as a clinical manifestation of this shared neurobiological pathway [36]. Therefore, we propose that the NERD-EPS overlap is not merely a co-occurrence of symptoms but a distinct DGBI subtype characterized by a primary somatosensory amplification mechanism, upon which psychological and social factors layer. This reconceptualization positions our findings not as a regional particularity but as a significant contribution to a global pathophysiological model, suggesting that targeted interventions addressing central sensitization and somatic symptom burden could be effective across diverse populations.

Although somatic symptoms and sleep disturbances are recognized manifestations of psychological distress [37–40]. their relative predominance in our cohort may reflect cultural influences. The Chinese norm of emotional restraint ("keeping anger and joy out of one's face") potentially favors expression of psychological distress through somatic complaints rather than direct verbalization, as often seen in Western populations [41,42]. This underscores that in Chinese healthcare settings, addressing somatic symptoms and sleep problems may be central to alleviating both psychological distress and gastrointestinal symptom severity.

An interesting finding specific to the NERD-EPS overlap phenotype concerns the association between sex and symptom severity. The fully adjusted multivariate model identified female sex as an independent factor associated with lower GSRS scores, a relationship not apparent in the univariate analysis. This suggests that the protective association of female sex was confounded in raw comparisons, which could occur if men in our tertiary care cohort delay help-seeking until reaching a higher overall burden. The observed protective effect may have a biological basis in sex-hormone interactions. For instance, testosterone has been reported to exert an inhibitory effect on the transient receptor potential vanilloid 1 channel, a key receptor mediating visceral hypersensitivity, in models of chronic inflammatory pain [43–45]. Thus, in the specific context of NERD-EPS overlap, such sex-specific mechanisms may contribute to the modest protective

association seen in women after accounting for key psychosocial confounders. High GSRS scores in younger patients, on the other hand, may be due to more sensitive pain perception than in older adults, more intense stress and emotions, etc., and in a similar situation, patients living in urban areas [46–51].

The strong, dose-dependent associations of somatic symptoms, sleep disturbances, and psychological distress with gastrointestinal symptom severity, as quantitatively demonstrated by our regression models, provide a clear evidence base for clinical management. Consequently, our findings advocate for integrated biopsychosocial assessment and suggest a structured screening and management pathway for NERD-EPS overlap patients. Based on the established hierarchy of effect sizes, we recommend routine administration of the PHQ-15 as a core screening tool for all suspected or confirmed NERD-EPS overlap cases in gastroenterology clinics. For patients identified with significant somatic or psychological burden, existing evidence from related conditions supports the potential utility of integrated approaches that combine acid-suppression therapy with cognitive-behavioral strategies targeting somatic symptoms (CBT-SS) or insomnia (CBT-I), as well as pharmacotherapies that address central sensitization (e.g., serotonin–norepinephrine reuptake inhibitors) [52–54]. Adjunctive approaches such as low-dose melatonin for sleep-gut rhythm regulation and stress management programs also represent rational candidates for future evaluation in this specific population [55–57]. This stepped biopsychosocial management strategy, guided by symptom severity assessment, may help identify patients who would benefit from a more integrated treatment approach beyond conventional pharmacotherapy alone. In current practice, a high PHQ-15 score should prompt clinicians to consider a multidisciplinary referral to integrated psychiatry or health psychology services, where available, to address the underlying central sensitization and somatic symptom burden.

## Strengths and limitations

This study used a multicenter cross-sectional design, covering tertiary hospitals in four provinces of China, and included a total of 800 patients with NERD and EPS overlap who met the Rome IV criteria, assessed multidimensional indicators using standardized scales such as the GSRS and the PHQ-15, and controlled for confounding factors through multiple regression analysis, with a geographically representative sample and sufficient statistical validity. However, it should be noted that a formal sample size calculation or power analysis was not performed a priori; the sample size of 800 was determined based on feasibility and the goal of recruiting a substantial multicenter cohort within the study period. The interaction analysis confirmed the independence of the associations, preventing the misattribution of effects that may arise from collinear variables and thereby allowing for a more precise discernment of core contributing factors. However, causality could not be determined in this study, biomarkers were not tested to elucidate the molecular mechanisms of somatic symptoms and visceral hypersensitivity, variables such as occupational classification may have affected the results, the generalizability of the results to the Chinese population needs to be verified in other populations. Although we adjusted for a range of demographic and clinical variables in our multivariable models, residual confounding from unmeasured or imperfectly measured factors (e.g., detailed socioeconomic status beyond education/residence, dietary habits, or complete medication adherence data) cannot be ruled out. Furthermore, the diagnosis of NERD-EPS overlap relied primarily on Rome IV criteria and symptom questionnaires. Objective physiological tests, such as 24-hour pH-impedance monitoring or gastric emptying studies, were not routinely performed in all participants due to the multicenter cross-sectional design and feasibility constraints. While this approach is consistent with many clinical and epidemiological studies of DGBI, it may introduce some heterogeneity in the patient phenotype and potential misclassification [4]. Beyond these considerations, this study focused specifically on the NERD-EPS overlap population to elucidate factors uniquely associated with symptom severity. While the absence of pure NERD or EPS control groups limits direct comparisons, our findings provide foundational data for future cohort studies to explore the temporal relationships and potential causal nature of these associatio. Finally, our sample was recruited exclusively from tertiary Traditional Chinese Medicine hospitals in China. This hospital-based recruitment strategy, within China's self-referral healthcare system, likely enriched our cohort with patients experiencing more severe symptoms and complex comorbidities. While this facilitated expert

diagnosis of FGIDs, it may limit the generalizability of our findings to community populations or primary care settings where symptom profiles are likely milder [58,59]. Moreover, cultural norms favoring somatization—where psychological distress is often expressed through physical symptoms—may further distinguish our cohort from populations in Western or primary care settings. Although these factors may limit the external applicability of prevalence estimates, they enhance the internal validity of the observed associations within this clinically salient group [40,41]. Notwithstanding these cultural considerations, the core biopsychosocial associations we identified, particularly the primacy of somatic symptoms, are likely to inform the understanding and management of NERD-EPS overlap in other healthcare settings, pending future validation.

## Conclusion

Symptom burden in NERD-EPS overlap is primarily driven by somatic symptoms, with significant contributions from psychological distress and sleep disturbances. We therefore recommend routine PHQ-15 screening in clinical practice to guide stepped biopsychosocial interventions. Future research should validate biomarkers for personalized treatment.

## Author contributions

**Conceptualization:** Mi Lv, Hui Che.

**Data curation:** Mi Lv, Hui Che, Zhaoxia Liu, Jinyi Xie, Fengyun Wang.

**Formal analysis:** Fengyun Wang.

**Funding acquisition:** Hui Che, Jinyi Xie, Fengyun Wang.

**Investigation:** Mi Lv, Hui Che, Jiayan Hu, Wenxi Yu, Zhaoxia Liu, Xiaolin Zhou, Binduo Zhou, Fengyun Wang.

**Methodology:** Mi Lv, Hui Che, Jiayan Hu, Wenxi Yu, Fengyun Wang.

**Project administration:** Fengyun Wang.

**Resources:** Fengyun Wang.

**Software:** Jiayan Hu, Wenxi Yu, Fengyun Wang.

**Supervision:** Hui Che, Jinyi Xie, Fengyun Wang.

**Validation:** Jinyi Xie, Fengyun Wang.

**Writing – original draft:** Mi Lv.

**Writing – review & editing:** Hui Che, Zhaoxia Liu, Xiaolin Zhou, Binduo Zhou, Jinyi Xie, Fengyun Wang.

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
