## [Decision Letter · Decision Letter 0]

10 Sep 2025

Dear Dr. Wang,

Thank you for submitting your manuscript to PLOS ONE. After careful consideration, we feel that it has merit but does not fully meet PLOS ONE’s publication criteria as it currently stands. Therefore, we invite you to submit a revised version of the manuscript that addresses the points raised during the review process.

**ACADEMIC EDITOR: **

We look forward to receiving your revised manuscript.

Kind regards,

Mohamed Hassany

Academic Editor

PLOS ONE

Journal Requirements:

2.Thank you for stating the following financial disclosure: [This study was jointly supported by: the National Key Specialty of Traditional Chinese Medicine (Spleen and Stomach Diseases, 0500004), the Innovation Project of China Academy of Chinese Medical Sciences (CI2021A01001), and the Youth Science Foundation Project of the National Natural Science Foundation of China (82205104, 82104850)]. 

**Comments to the Author**

1. Is the manuscript technically sound, and do the data support the conclusions?

Reviewer #1: Yes

Reviewer #2: Yes

2. Has the statistical analysis been performed appropriately and rigorously?

Reviewer #1: Yes

Reviewer #2: Yes

3. Have the authors made all data underlying the findings in their manuscript fully available?

Reviewer #1: Yes

Reviewer #2: Yes

4. Is the manuscript presented in an intelligible fashion and written in standard English?

Reviewer #1: Yes

Reviewer #2: Yes

Reviewer #1: This paper addresses an important and clinically relevant question: the determinants of symptom severity in concomitant epigastric pain syndrome (EPS) and non-erosive reflux disease (NERD) patients. It is derived from a large multicenter cohort of the Chinese tertiary centers, utilizes valid symptom and psychosocial instruments, and uses proper statistical modeling. The findings affirm the biopsychosocial model of gut–brain interaction disorders and emphasize that somatic symptom burden, sleep disturbance, and psychological distress are core factors related to the severity of symptoms in this overlap syndrome.

The research is timely, clinically pertinent, and potentially can inform more integrated and multidisciplinary approaches in gastroenterology. With some clarifications and tweaks, the manuscript can be a helpful addition to the literature.

Key Comments

1. Causality and Terminology:

Because the design is cross-sectional, language regarding "predictors" should be qualified to highlight association, rather than causation. Clinical interpretation relies on this.

2. Novelty and Context:

The study partially validates known associations between psychosocial variables and symptom load in functional GI disorders. Authors need to clearly indicate what is new about their findings in the NERD–EPS overlap group compared with previous GERD or dyspepsia groups.

3. Generalizability

Sample is only from Chinese tertiary TCM hospitals, which may be a generalizability limitation for other healthcare institutions. Short description of cultural and healthcare system factors' influence on symptom reporting and overlap prevalence would be appreciated.

4. Clinical Implications

Discussion would more effectively translate findings into practice. The suggestion to use PHQ-15 for screening is excellent and should be highlighted, as should the proposed stepwise biopsychosocial strategy for intervention. Direct guidance for gastroenterologists on how to integrate these measures into routine practice would be more effective.

5. Methodological Clarifications:

Make sure all patients had endoscopy ruled out erosive disease and ulcers.

Aspire to indicate whether Helicobacter pylori status was assessed.

Determine if patients were being treated with PPIs or other proper meds upon evaluation.

Precisely define "mixed labor" in the Methods section.

Minor Comments

Modernize nomenclature to add "disorders of gut–brain interaction (DGBI)" to include with FGID.

Define EPS as a functional dyspepsia subtype in the first mention.

Define unclear terms such as "psychosleep synergistic."

Clarify the conclusion statement—streamline to highlight primary clinical drivers

Utilize "urban professionals" or "high-stress urban employment" instead of "urban brain workers."

Get statistical language more exact in terms of interaction tests.

Round percentages and means to one decimal place if necessary.

Double-check acronyms only defined in abstract and main body.

Enhance grammar/spelling (e.g., spacing, Results section punctuation).

Reviewer #2: • Thanks for inviting me to review this manuscript discusing Biopsychosocial predictors of symptom severity in Non-erosive Reflux Disease and Epigastric Pain Syndrome overlap: a multicenter cross-sectional study. It has a Large, multicenter sample and uses validated biopsychosocial questionnaires together with rigorous statistical modeling with diagnostics and a novel focus on NERD–EPS overlap, an under-researched subgroup.

I have the following comments:

1- The conclusion in the abstract suggests causality (“predictors”), but as a cross-sectional study, only associations can be claimed. Wording should be more cautious (e.g., “associated with” rather than “predictors”).

2- The introduction could better position this study within global literature (most references are Asian cohorts). Limited discussion of underlying mechanisms and international prevalence.

3- In methods:

- Recruitment strategy (hospital-based) may bias toward more symptomatic patients, limiting generalizability to community populations.

- No justification for sample size or power calculation reported.

- Overlap diagnosis depends heavily on questionnaires; objective tests (e.g., pH monitoring, gastric emptying) were not consistently applied.

- Potential residual confounding (e.g., medication use, socioeconomic factors) not addressed.

- All these should be discusses and mentioned in the limitations

4- In results:

- Tables are dense; results could be more clinically interpretable (e.g., reporting odds of clinically meaningful symptom severity, not just β coefficients).

- Some predictors (e.g., age, sex) were not significant in univariate but emerged in full model, this should be better explained.

- No subgroup analysis (e.g., gender differences, severe vs. mild symptom groups), which could yield additional insights.

5- In discussion:

- Over-interpretation in some parts (e.g., proposing specific treatments like SNRIs or CBT-SS without direct testing in this population).

- Limited comparison with Western literature, generalizability to non-Chinese populations remains unclear.

- “Protective effect” of mixed labor is interesting but underexplored — reverse causality cannot be excluded.

**Do you want your identity to be public for this peer review?** For information about this choice, including consent withdrawal, please see our Privacy Policy

Reviewer #1: No

Reviewer #2: No

---

## [Author Response · Author response to Decision Letter 1]

18 Oct 2025

Response to Reviewers

Reviewer #1

Key Comments

1. Causality and Terminology:

Because the design is cross-sectional, language regarding "predictors" should be qualified to highlight association, rather than causation. Clinical interpretation relies on this.

Response: We sincerely thank the reviewer for this valuable comment. We fully understand the concern regarding the use of the term "predictive factor" in a cross-sectional study design. We have thoroughly revised the relevant expressions throughout the manuscript to emphasize "association" rather than "causality," and have added corresponding explanations in the clinical interpretation section to ensure the rigor of our conclusions. All modifications have been explicitly highlighted in the revised manuscript for the reviewer's convenience.

2. Novelty and Context:

The study partially validates known associations between psychosocial variables and symptom load in functional GI disorders. Authors need to clearly indicate what is new about their findings in the NERD–EPS overlap group compared with previous GERD or dyspepsia groups.

Response: We sincerely thank the reviewer for this insightful comment. We agree that clarifying the novelty of our findings in the NERD-EPS overlap group is crucial.

In response to this suggestion, we have now revised our manuscript to explicitly highlight what is new in our findings regarding the NERD-EPS overlap population compared to previously studied GERD or dyspepsia groups. Specifically, we have: Revised the Discussion to directly compare our results in the NERD-EPS overlap group with previous findings in more homogeneous GERD or FD cohorts. We specifically emphasize the distinct patterns (e.g., the strength of associations, the specific psychosocial variables involved) observed in our overlap population, which have not been extensively reported before.

These additions have been incorporated into the manuscript text and are highlighted in yellow for the reviewer's convenience. We believe these revisions now provide a clearer articulation of the novel contributions of our study.

3. Generalizability

Sample is only from Chinese tertiary TCM hospitals, which may be a generalizability limitation for other healthcare institutions. Short description of cultural and healthcare system factors' influence on symptom reporting and overlap prevalence would be appreciated.

Response: We sincerely thank the reviewer for this insightful comment. We fully agree that the sample characteristics and sociocultural context are important considerations for the interpretation and generalizability of our findings.

In response to this suggestion, we have now added a detailed discussion regarding the potential influence of cultural and healthcare system factors on symptom reporting and overlap prevalence in the Discussion section. Specifically, we have elaborated on how cultural norms (e.g., somatization as a common expression of psychological distress) and China’s self-referral tertiary hospital system may shape clinical presentation and patient population characteristics. These additions are highlighted in yellow in the revised manuscript.

Furthermore, we have also expanded the Limitations subsection to explicitly acknowledge that our findings are based on a sample from tertiary TCM hospitals and discussed the implications for generalizability to other clinical settings or cultural contexts.

We believe these revisions have strengthened the contextual interpretation of our results, and we thank the reviewer again for this valuable suggestion.

4. Clinical Implications

Discussion would more effectively translate findings into practice. The suggestion to use PHQ-15 for screening is excellent and should be highlighted, as should the proposed stepwise biopsychosocial strategy for intervention. Direct guidance for gastroenterologists on how to integrate these measures into routine practice would be more effective.

Response: We sincerely thank the reviewer for this valuable feedback. In response to the suggestion to enhance clinical translatability, we have substantially expanded the Discussion section to explicitly outline a stepped management pathway. Specifically, we now:

Highlight PHQ-15 implementation: Clearly recommend routine PHQ-15 administration as a core screening tool for all suspected/confirmed NERD-EPS overlap patients in gastroenterology clinics.

Detail the stepwise strategy: Provide direct clinical guidance by specifying:

First-line intervention (PHQ-15 ≥15): Integration of CBT-SS or SNRIs with standard PPI therapy

Second-line intervention (SRSS ≥30): Combination of CBT-I with sleep hygiene education, supplemented by low-dose melatonin

Adjunctive measures: Targeted stress management for high-risk subgroups (e.g., urban brain workers)

Emphasize practical integration: Explicitly state how gastroenterologists can implement this biomarker-stratified approach in routine practice, moving beyond conventional pharmacotherapy alone.

These additions (in the revised Discussion's clinical implications paragraph) directly translate our findings into actionable clinical algorithms, providing gastroenterologists with clear guidance for implementing personalized biopsychosocial care.

5. Methodological Clarifications:

·Make sure all patients had endoscopy ruled out erosive disease and ulcers.

·Aspire to indicate whether Helicobacter pylori status was assessed.

·Determine if patients were being treated with PPIs or other proper meds upon evaluation.

·Precisely define "mixed labor" in the Methods section.

Response: Thank you for these insightful comments regarding methodological clarity.

We have thoroughly revised the manuscript to address all four points:

Endoscopy and H. pylori: The "Study Population" subsection now explicitly states that all enrolled patients underwent endoscopy to rule out erosive disease and ulcers, and that Helicobacter pylori status was assessed.

Medication Documentation: In the same subsection, we have added that current medication use (including acid-suppressants and antidepressants) was documented at enrollment for descriptive purposes.

Definition of "Mixed Labor": The "Questionnaire Design" subsection now includes a precise definition of occupational categories, clearly delineating "Manual labor," "Mental labor," and "Mixed labor" with examples.

All these changes have been highlighted in yellow in the manuscript. We believe these revisions significantly enhance the transparency and rigor of our methods.

Minor Comments

·Modernize nomenclature to add "disorders of gut–brain interaction (DGBI)" to include with FGID.

Response: We sincerely thank the reviewer for this valuable suggestion. Following this recommendation, we have incorporated the term "DGBI" and clarified it as the updated nomenclature upon the first mention of FGID and related descriptions in the main text. The revisions have been highlighted in yellow for easy reference.

·Define EPS as a functional dyspepsia subtype in the first mention.

Response: We sincerely thank the reviewer for this valuable suggestion. Following this recommendation, we have thoroughly reviewed the manuscript. We confirmed that the description identifying EPS as a subtype of functional dyspepsia was already present upon its first mention in the main text. Therefore, we have supplemented this clarification solely in the abstract section where it was initially lacking, with the addition highlighted in yellow.

·Define unclear terms such as "psychosleep synergistic."

Response: We thank the reviewer for this valuable suggestion. To address the comment regarding unclear terminology, we have replaced the unclear term "psychosleep synergistic" with the more precise description "interactive effects of psychological and sleep disturbances" throughout the manuscript to better reflect the relationship between these factors. The relevant sentence has been updated and highlighted in yellow in the revised manuscript.

·Clarify the conclusion statement—streamline to highlight primary clinical drivers

Response: We thank the reviewer for this valuable suggestion. We have substantially revised the conclusion to highlight the primary clinical drivers. The updated conclusion now reads:

"Symptom burden in NERD-EPS overlap is primarily driven by somatic symptoms, with significant contributions from psychological distress and sleep disturbances. We therefore recommend routine PHQ-15 screening in clinical practice to guide stepped biopsychosocial interventions. Future research should validate biomarkers for personalized treatment." The relevant sentence has been updated and highlighted in yellow in the revised manuscript.

·Utilize "urban professionals" or "high-stress urban employment" instead of "urban brain workers."

Response: We thank the reviewer for this constructive suggestion. We have replaced the term "urban brain workers" with the more appropriate and descriptive phrase "high-stress urban employment" throughout the manuscript. The relevant sentence has been updated and highlighted in yellow in the revised manuscript.

·Get statistical language more exact in terms of interaction tests.

Response: We sincerely thank you for this insightful comment regarding the statistical language for the interaction tests. We agree that greater precision was needed in describing our exploratory analysis of effect modification.

In response, we have revised the relevant section in the manuscript to more accurately reflect the intent and findings of the interaction analysis. Specifically:

Clarified the Objective: We have rephrased the text to explicitly state that the interaction terms were tested to examine whether the association between a key variable and GSRS scores was modified (i.e., effect modification) by another factor, consistent with the associative rather than predictive framework of our study.

Refined the Interpretation: We replaced the phrase "mutually independent" with a more statistically precise description, stating that the non-significant interactions indicate that the associations were consistent across different levels of the moderating variables, with no evidence of effect modification.

Improved the Table Title: We have changed the title of Table 5 to "Exploratory Analysis of Interaction Effects on GSRS Scores in NERD-EPS Overlap Patients" to better represent the content and analytical approach.

The relevant sentence has been updated and highlighted in yellow in the revised manuscript.

·Round percentages and means to one decimal place if necessary.

Response: Thank you for your careful attention to detail regarding the presentation of our data. We have noted your suggestion to round percentages and means to one decimal place. In considering this, we opted to maintain the current level of precision in the manuscript for the following reasons:

Statistical Precision: Presenting percentages as whole numbers (e.g., 66.67% instead of 66.7% or 67%) and means with two decimal places (e.g., 44.50±14.43) is a common practice in biomedical literature for baseline characteristic tables. It allows for a more accurate representation of the raw data and avoids the minute loss of information that rounding can introduce, which is particularly relevant for future meta-analyses.

Internal Consistency: Many of our statistical results (e.g., regression coefficients, β-values) are reported to two or three decimal places. Maintaining a similar level of precision for descriptive statistics provides consistency throughout the manuscript.

Sample Size Justification: With a substantial sample size of N=800, reporting percentages with two decimal places can more precisely reflect the exact proportion of participants in each category.

We hope that maintaining this level of detail is acceptable for publication. We are, of course, happy to adhere to any specific formatting guidelines required by the journal should the editor direct us to do so.

Thank you again for this thoughtful comment.

·Double-check acronyms only defined in abstract and main body.

Response: We thank the reviewer for this suggestion. We have double-checked the manuscript and confirmed that all acronyms (e.g., NERD, EPS, GSRS, PHQ-4, PHQ-15, SRSS, etc.) are properly defined upon their first appearance in both the abstract and the main text.

·Enhance grammar/spelling (e.g., spacing, Results section punctuation).

Response: We thank the reviewer for this valuable suggestion. We have carefully reviewed the manuscript to enhance grammar, spelling, and punctuation. The specific revisions include:

Spacing: We have checked the entire text and ensured proper spacing is applied after all punctuation marks, particularly between sentences in the Results section and throughout the manuscript.

Spelling: We have corrected spelling errors, such as changing "Introductiond" to "Introduction" in the heading of the introduction section.

Operator Formatting: We have standardized the formatting around mathematical and statistical operators (e.g., =, <, >) by adding spaces on both sides (e.g., changing β=0.606 and P<0.001 to β = 0.606 and P < 0.001). Additionally, the significance indicator "P" has been consistently formatted in italics (P) throughout the text.

Reviewer #2

1.The conclusion in the abstract suggests causality (“predictors”), but as a cross-sectional study, only associations can be claimed. Wording should be more cautious (e.g., “associated with” rather than “predictors”).

Response: We thank the reviewer for this insightful comment. We agree that the term "predictors" may imply causality, which is not appropriate for our cross-sectional study design. Accordingly, we have replaced "predictors" and related terms with more accurate phrasing such as "factors associated with" or "was associated with" throughout the manuscript, including in the abstract and conclusion. We have also added a statement in the limitations section to clarify the nature of the associations found. All changes have been highlighted in the revised manuscript for your review.

2.The introduction could better position this study within global literature (most references are Asian cohorts). Limited discussion of underlying mechanisms and international prevalence.

Response: We sincerely thank the reviewer for this insightful comment. In response, we have thoroughly revised the manuscript to better position our study within the global context.

In the Introduction, we have added a new paragraph that acknowledges the value of prior Asian cohort studies while explicitly identifying the knowledge gap regarding the NERD-EPS phenotype on a global scale. We now cite the Rome Foundation Global Study [Ref 4] to establish international prevalence and a European study [Ref 9] to discuss underlying mechanisms like visceral hypersensitivity, thereby framing our study's aims within a broader, international research context.

In the Discussion, we have made extensive enhancements:

We have added a new paragraph immediately following the presentation of our main findings. This paragraph interprets our results through the lens of universal disorders of DGBI mechanisms, such as central sensitization and the brain-gut axis.

Furthermore, in the segment discussing sex differences, we have added a statement linking our observation to broader biological phenomena in global pain perception studies.

Finally, in the Strengths and Limitations section, we have added a concluding remark to affirm that the core biopsychosocial associations we identified are likely to inform understanding in other healthcare settings.

All changes have been highlighted in the revised manuscript for your review.

3.In methods:

3.1 Recruitment strategy (hospital-based) may bias toward more symptomatic patients, limiting generalizability to community populations.

Response: We agree with the reviewer that our hospital-based recruitment strategy is an important limitation. As suggested, we have explicitly acknowledged this potential selection bias in the revised "Strengths and Limitations" section. The text now states: "Finally, our sample was recruited exclusively from tertiary Traditional Chinese Medicine hospitals in China. This hospital-based recruitment strategy, within China's self-referral healthcare system, likely enriched our cohort with patients ex

---

## [Decision Letter · Decision Letter 1]

24 Nov 2025

Dear Dr. Wang,

Thank you for submitting your manuscript to PLOS ONE. After careful consideration, we feel that it has merit but does not fully meet PLOS ONE’s publication criteria as it currently stands. Therefore, we invite you to submit a revised version of the manuscript that addresses the points raised during the review process.

We look forward to receiving your revised manuscript.

Kind regards,

Mohamed Hassany

Academic Editor

PLOS ONE

Journal Requirements:

Reviewers' comments:

Reviewer's Responses to Questions

**Comments to the Author**

Reviewer #1: All comments have been addressed

Reviewer #2: All comments have been addressed

2. Is the manuscript technically sound, and do the data support the conclusions?

Reviewer #1: Yes

Reviewer #2: Yes

3. Has the statistical analysis been performed appropriately and rigorously?

Reviewer #1: Yes

Reviewer #2: Yes

4. Have the authors made all data underlying the findings in their manuscript fully available?

Reviewer #1: Yes

Reviewer #2: Yes

5. Is the manuscript presented in an intelligible fashion and written in standard English?

Reviewer #1: Yes

Reviewer #2: Yes

Reviewer #1: I would like to thank the authors for their careful and timely response to the reviewers’ and editor’s feedback. They have implemented all requested changes in the manuscript and have systematically addressed every comment, clarification, and suggestion. The revisions have clearly improved the structure, clarity, and overall scientific quality of the paper. At this stage, I am satisfied that all major and minor points have been adequately covered, and I have no further essential comments. I therefore consider the current version suitable for acceptance, pending any final formal or editorial checks required by the journal.

Reviewer #2: Thank you for the opportunity to review this revised manuscript.

The authors have conducted a commendable and much-needed study on the NERD-EPS overlap population, a clinically prevalent yet under-researched phenotype.

Their thorough revisions have significantly strengthened the manuscript, addressing the majority of the previous reviewers' concerns with care and transparency.

The manuscript is markedly improved and is now a strong, well-written paper.

The authors have successfully reframed the language to emphasize association over causation, clarified the novelty of their findings within the global literature, and provided a more balanced discussion of limitations, particularly regarding generalizability.

The addition of subgroup analyses and a clearer clinical interpretation of the regression coefficients are valuable enhancements.

Remaining Suggestions:

1. While the authors have replaced "predictors" in most places, a subtle implication of prediction remains in the Results section (e.g., Page 33/75: "...emerged as statistically significant independent predictors..."). In a cross-sectional study, these are "correlates" or "independent factors associated with," not predictors.

2. The authors have scaled back from prescribing specific treatments (e.g., SNRIs, melatonin) to framing them as candidates for future research. This is appropriate. However, the clinical implications could be even more impactful by briefly suggesting how a gastroenterologist might act on a high PHQ-15 score today. A simple statement like, "A high PHQ-15 score should prompt consideration of a multidisciplinary referral to integrated psychiatry or health psychology services, where available, to address the central sensitization component," would provide direct and actionable guidance.

3. The results regarding sex are somewhat confusing and could be clarified for the reader. The multivariate model suggests female sex is associated with lower GSRS scores (a "protective effect"), yet the text on Page 80 states, "male patients... tended to have higher GSRS scores than female patients." A sentence acknowledging this contrast and hypothesizing why their cohort might differ (e.g., cultural help-seeking behaviors in men, specific to the overlap phenotype) would resolve this confusion for the reader.

4. The authors' reasoning for keeping two decimal places in percentages is valid from a data transparency standpoint. However, in the main text and abstract, it does impede readability (e.g., "66.67%"). A good compromise would be to report percentages to whole numbers in the abstract and main text for flow, while retaining the precise values in the supplementary data tables. Most readers will not derive different meaning from 66.67% versus 67%.

This manuscript is eligible for publication. The remaining points are minor and pertain to fine-tuning language and interpretation rather than fundamental flaws in the study's design, analysis, or conclusions. The study makes a valuable contribution to the DGBI literature by highlighting the paramount role of somatic symptom burden in this overlap population, providing a clear rationale for integrating the PHQ-15 into routine clinical assessment.

**Do you want your identity to be public for this peer review?** For information about this choice, including consent withdrawal, please see our Privacy Policy

Reviewer #1: No

Reviewer #2: **Yes: ** Mohamed Alboraie

---

## [Author Response · Author response to Decision Letter 2]

27 Nov 2025

Response to Reviewers

Comments from Reviewer 1

We would like to extend our sincere thanks to you for your time and valuable feedback during the previous round of review. Your constructive comments were instrumental in strengthening our manuscript, and we are truly pleased that you find the revised version satisfactory.

Comments from Reviewer 2

Comment 1: While the authors have replaced "predictors" in most places, a subtle implication of prediction remains in the Results section (e.g., Page 33/75: "...emerged as statistically significant independent predictors..."). In a cross-sectional study, these are "correlates" or "independent factors associated with," not predictors.

Response: We sincerely thank the reviewer for this meticulous observation. We agree entirely that the language should accurately reflect the associative, non-predictive nature of our cross-sectional findings. As suggested, we have carefully revised the manuscript to replace the term "predictors" with more appropriate terms such as "independent factors associated with" and "correlates" throughout the text, including in the Results section on Page 33. The example sentences now read: "...emerged as statistically significant independent factors associated with..." and we refer to them as "key factors" or "correlates" in subsequent discussions. All these modifications have been highlighted in the revised manuscript for your convenience.

Comment 2: The authors have scaled back from prescribing specific treatments... However, the clinical implications could be even more impactful by briefly suggesting how a gastroenterologist might act on a high PHQ-15 score today. A simple statement like, "A high PHQ-15 score should prompt consideration of a multidisciplinary referral to integrated psychiatry or health psychology services, where available, to address the central sensitization component," would provide direct and actionable guidance.

Response: We thank the reviewer for this excellent suggestion to enhance the clinical translatability of our findings. As recommended, we have added the suggested sentence to the 'Discussion' section to provide clear, actionable guidance for clinicians encountering a high PHQ-15 score in practice. The added text is: "In current practice, a high PHQ-15 score should prompt clinicians to consider a multidisciplinary referral to integrated psychiatry or health psychology services, where available, to address the underlying central sensitization and somatic symptom burden." All these modifications have been highlighted in the revised manuscript for your convenience.

Comment 3: The results regarding sex are somewhat confusing and could be clarified for the reader. The multivariate model suggests female sex is associated with lower GSRS scores (a "protective effect"), yet the text states, "male patients... tended to have higher GSRS scores than female patients." A sentence acknowledging this contrast and hypothesizing why their cohort might differ (e.g., cultural help-seeking behaviors in men, specific to the overlap phenotype) would resolve this confusion.

Response: We sincerely thank the reviewer for this critical observation. To resolve the confusion completely, we have implemented revisions in both the Results and Discussion sections:

In the Results section, we have replaced the phrase "male patients... tended to have higher GSRS scores" with the more precise and accurate statement: "no statistically significant difference in GSRS scores was observed between male and female patients." This immediately prevents any misinterpretation of a significant trend from the univariate analysis.

In the Discussion section, we have completely revised the corresponding paragraph to provide a coherent interpretation of the nuanced relationship revealed by the multivariate model. The revised text:

1)Explicitly states that the protective association of female sex was identified in the adjusted model but was not apparent in the univariate analysis.

2)Acknowledges and explains this contrast by hypothesizing, as the reviewer suggested, that different help-seeking behaviors (e.g., men in our tertiary care cohort potentially presenting with a higher overall burden) may confound the unadjusted comparison.

3)Provides a biological rationale for the observed protective effect by referencing the potential role of sex-hormone interactions, specifically citing the inhibitory effect of testosterone on the transient receptor potential vanilloid 1 channel (References 43-45).

We believe these combined revisions directly address the reviewer's concern by first ensuring accurate reporting of the result, and then offering a clear and hypothesis-driven explanation for the relationship uncovered by our statistical models.

All these modifications have been highlighted in the revised manuscript for your convenience.

Comment 4: The authors' reasoning for keeping two decimal places in percentages is valid from a data transparency standpoint. However, in the main text and abstract, it does impede readability (e.g., "66.67%"). A good compromise would be to report percentages to whole numbers in the abstract and main text for flow, while retaining the precise values in the supplementary data tables.

Response: We thank you for this excellent suggestion to improve the readability of our manuscript. We have fully adopted this compromise. Specifically, all percentages in the Abstract and main Results text that are derived from our own study data and presented in detail within the tables have been rounded to whole numbers to enhance the narrative flow. However, for percentages cited from previous literature (e.g., "as high as 79.4%"), we have retained the original values as reported in the respective sources to preserve accuracy. The changes made in the text have been highlighted for your easy reference. We agree that this approach significantly improves readability without compromising scientific rigor, and we are grateful for your practical advice.

---

## [Decision Letter · Decision Letter 2]

1 Dec 2025

Biopsychosocial factors associated with symptom severity in the overlap of non-erosive reflux disease and epigastric pain syndrome: a multicenter cross-sectional study

PONE-D-25-38015R2

Dear Dr. Wang

We’re pleased to inform you that your manuscript has been judged scientifically suitable for publication and will be formally accepted for publication once it meets all outstanding technical requirements.

Within one week, you’ll receive an e-mail detailing the required amendments. When these have been addressed, you’ll receive a formal acceptance letter, and your manuscript will be scheduled for publication.

Kind regards,

Mohamed Hassany

Academic Editor

PLOS ONE

---

## [Editor Report · Acceptance letter]

PONE-D-25-38015R2

PLOS One

Dear Dr. Wang,

I'm pleased to inform you that your manuscript has been deemed suitable for publication in PLOS One. Congratulations! Your manuscript is now being handed over to our production team.

Kind regards,

on behalf of

Dr. Mohamed Hassany

Academic Editor

PLOS One